# COVID-19-related institutional betrayal associated with trauma symptoms among undergraduate students

**Alexis A. Adams-Clark**[1,2]*, **Jennifer J. Freyd**[1,2,3]

**1** Department of Psychology, University of Oregon, Eugene, Oregon, United States of America, **2** Center for Institutional Courage, Inc., Palo Alto, California, United States of America, **3** Department of Psychiatry and Behavioral Sciences, Stanford University School of Medicine, Stanford, California, United States of America

* aadamscl@uoregon.edu

**Data Availability Statement:** Data is publicly available at: Study 1: https://mfr.osf.io/render?url=https://osf.io/yxhc7/?direct%26mode=render%26action=download%26mode=render Study 2: https://mfr.osf.io/render?url=https://osf.io/nrejq/?

## Abstract

Individuals are dependent on institutions (e.g., universities, governments, healthcare systems) to protect their safety and advocate for their needs. When institutions harm the individuals who depend on them, they commit *institutional betrayal*, which has been associated with numerous negative outcomes in prior research. Throughout the COVID-19 pandemic, students have entrusted universities to protect both their health and their educational opportunities. However, many universities have failed to meet these expectations, and it is likely that many students experience COVID-19-related institutional betrayal. In two similar studies, we examined the prevalence and correlates of institutional betrayal among undergraduate students at a large, public university in the Northwest United States during the fall 2020 and winter 2021 quarters. In both studies, more than half of students endorsed at least one type of COVID-19-related institutional betrayal, and higher institutional betrayal ratings were significantly correlated with both current trauma symptoms and COVID-19-related avoidance and intrusion cognitions. In Study 2, the relationship between COVID-19-related institutional betrayal and current trauma symptoms remained significant, even when controlling for gender, personal and familial COVID-19 infection, and past trauma history. These results indicate that COVID-19 institutional betrayal is common and may be uniquely associated with distress among undergraduate students. We suggest it would behoove university institutions to reduce COVID-19-related institutional betrayal.

## Introduction

*In the absence of any national strategy for tackling the coronavirus pandemic, colleges and universities in the United States are on their own when it comes to deciding whether and how to bring students back for the autumn term, which has already started for some institutions. Many are relying on their own experts, resulting in a wide range of approaches. . .It all amounts to a gigantic, unorganized public-health experiment—with millions of students and an untold number of faculty members and staff as participants.*

[1 p510–511]

direct%26mode=render%26action=download%26mode=render The OSF project is identified with the following DOI: 10.17605/OSF.IO/M57PT.

**Funding:** The author(s) received no specific funding for this work.

**Competing interests:** The authors have declared that no competing interests exist.

Individuals are frequently dependent on societal institutions (e.g., universities, governments, healthcare systems) to protect their safety, provide them with vital services, and advocate for their needs. There are few times in recent history when this has been truer than during the coronavirus pandemic; in such a time of crisis, individuals turned to various institutions to enact and enforce policies to curb the spread of COVID-19, mobilize efforts to create treatments and vaccines, and equitably distribute care to those infected. However, in multiple domains, institutional efforts were left wanting, as COVID-19 infections continued to proliferate.

The term *institutional betrayal* [2, 3] can be used to describe such an occurrence. Institutional betrayal manifests when an institution fails to fulfill its obligations to institutional members who entrust and depend upon it. Such a betrayal can occur through both institutional actions (e.g., an institution actively committing a transgression or violation against a member) or inactions (e.g., an institution failing to enact appropriate policies or respond adequately to an expressed concern). In past research, institutional betrayal has largely been studied in the context of campus sexual assault [4], healthcare experiences [5], university study abroad programs [6], and military sexual trauma [7], but it may also apply to a range of experiences throughout the COVID-19 pandemic.

Although scholars have identified several instances of institutional betrayal occurring throughout the COVID-19 pandemic, such as betrayal by healthcare systems [8] and by government leaders [9], less commentary exists regarding possible experiences of institutional betrayal by college students in the context of their university institution. Throughout the COVID-19 pandemic, students have entrusted, and at times forcibly made to rely upon, universities to protect both their health and educational opportunities, even as COVID-19 cases rise on many campuses [10]. Although many universities, including the authors' own institution, have enacted numerous policies that aim to curb the spread of COVID-19 (e.g., mask mandates, frequent cleaning, social distancing), many have also simultaneously created situations in which COVID-19 transmission is more likely (e.g., holding some in-person classes, requiring first-year students to live in on-campus dormitories with limited exceptions), which have led to rising rates of infection on campuses. Such institutions commit several types of institutional betrayal studied in prior literature, such as the creation of an environment in which threats to the safety of institutional members seem common or inconsequential [4]. Even universities that have implemented remote-only instruction may risk committing other common types of institutional betrayal, including the creation of an environment where continued membership is difficult [4], either due to remote learning, financial, or personal challenges.

COVID-19-related institutional betrayal among college students is particularly important to acknowledge and measure, given that prior research suggests that specific harm is created when trusted institutions fail to fulfill promises. Experiences of institutional betrayal in other contexts have been found to be associated with numerous negative mental health outcomes, including trauma symptoms [4, 11], physical health outcomes [12], suicidal ideation [7], and disengagement from healthcare services [5]. If students experience COVID-19-related institutional betrayal, they may be also be experience similar outcomes. The harms of COVID-19-related institutional betrayal may be particularly toxic, as institutional betrayal likely compounds the existing stress that college students have been experiencing since the pandemic began [13]. Notably, initial research on college students' experiences during COVID-19 found a significant correlation between psychological symptoms and trust in the government's management of the pandemic [14]. Such a pattern may similarly exist among psychological symptoms and trust in the university's management of the pandemic.

The current studies sought to describe and characterize the prevalence of students' experiences of COVID-19-related institutional betrayal and their relationship with trauma-related

outcomes. Collecting data from two samples of undergraduate students in the fall 2020 (Study 1) and winter 2020 (Study 2) academic quarters, we measured prevalence rates of 12 types of COVID-19-related institutional betrayal, as well as individual factors that may predict experiences of institutional betrayal, including living on campus, living in the university town, taking in-person classes, and degree of experience with COVID-19 infection. Students who are required to live on campus, live within the university town, and/or take in-person classes may have greater day-to-day contact with the university institution and are more likely to be affected by its policies; thus, they may be more at risk for institutional betrayal. Similarly, those who either know someone who has contracted COVID-19 (Study 1) or contracted COVID-19 themselves (Study 2) may also report greater institutional betrayal, particularly if this infection occurred within the context of the university institution. In addition to these variables, we measured students' self-reported general trauma symptoms, an outcome that has been commonly studied in prior studies of institutional betrayal. Finally, we measured COVID-19-specific avoidance and intrusion cognitions as a potential trauma-related outcome associated with institutional betrayal not captured by a general measure of trauma symptoms. Whereas general trauma symptoms may include nonspecific symptoms of anxiety, depression, and sleep problems, avoidant and intrusion cognitions are frequently related to a specific event or topic, such as a global pandemic. Symptoms of intrusion (i.e., unwanted thoughts/images of an event) and avoidance (i.e., efforts to reduce contact with thoughts or reminders of the event) commonly associated with Post-Traumatic Stress Disorder [15]. Both may be associated with institutional betrayal.

## Study 1

Study 1 served as an initial exploration of the relationship between COVID-19-related institutional betrayal. We began with a descriptive question and two main hypotheses. We had three main research goals.

1. To examine the prevalence of COVID-19-related institutional betrayal experiences among undergraduates by describing the number of students experiencing at least one type of institutional betrayal, as well as each particular type of institutional betrayal.

2. Test the hypothesis that COVID-19-related institutional betrayal ratings would be higher among those students living on campus, living in the university town, taking in-person classes, and who report knowing someone who was infected with COVID-19, compared to their peers not similarly situated.

3. Test the hypothesis that experiences of COVID-19-related institutional betrayal would be associated with both current trauma symptoms and COVID-19-specific intrusion and avoidance cognitions, and that these associations would persist, even when controlling for gender and knowing someone who was infected with COVID-19.

## Study 1 method

### Participants

Participants were recruited from the Human Subjects Pool at a large, public university in the Northwest United States. The university's Human Subjects Pool contains undergraduate students currently enrolled in introductory undergraduate psychology and linguistics courses, and these students receive course credit for their participation in research studies. Students in the Human Subjects pool are not aware of the topic of any given study prior to signing up,

which reduces self-selection bias (although they do have the option to end participation during the informed consent process or at any time throughout the study). This study was part of a larger data collection project, which included other measures of individual differences (e.g., personality characteristics, social attitudes) that were not examined for this study's research questions.

A total of 346 undergraduate students signed up and consented to participate in Study 1. Participants who failed to correctly answer at least four out of five "attention check" questions randomly located throughout the survey (e.g., "please choose 'strongly agree' if you are paying attention") were removed prior to data analysis ($n = 37$). The final sample used for analysis consisted of 309 participants (71.5% women, 26.5% men, 1.9% non-binary/gender-nonconforming). The majority of participants were White (75.4%) and first-year (44.3%) or second-year (29.4%) students. The average age of participants was 19.39 ($SD = 1.45$). Although women were over-represented (as is typical in psychology courses and of our Human Subject Pool drawn from such courses), other demographic characteristics of the sample were approximately representative of the university. According to official university statistics, approximately 55.3% of the university student body is female (44.7% male), and 60.0% of students are both white and non-Hispanic/Latino (not including 2.1% of students whose race and ethnicity are unknown and 8.1% of students who are international students). The average age of undergraduate students is 20.9 [16]. Full demographic characteristics for Study 1 are listed in Table 1A.

Study 1 data were collected during the fall 2020 quarter of the academic year (October 26—December 4, 2020), during which COVID-19 infections were steadily climbing at the university, local, and national level. The university at the center of the current investigation runs on a three-quarter academic year (with an optional fourth summer quarter) and adopted a primarily remote learning environment for 2020–2021. However, the university required all first-year students to live in dormitories on campus (with a few exceptions), and a minority of classes were held in person.

## Measures

**COVID-19-related institutional betrayal.** COVID-19-related institutional betrayal was measured using an adapted version of the Institutional Betrayal Questionnaire-12 (IBQ-12) [4, 12]. The IBQ consists of 12 items listing actions or inactions by an institution in response to a traumatic event, and it has been established as a valid measure of institutional betrayal. Although originally designed to assess universities' responses to instances of sexual violence, the measure was adapted to apply to universities' responses to the COVID-19 pandemic. Participants were instructed to answer each item by selecting "Yes," "No," or "Not Applicable." In addition to reading the original item text, examples of how these items may apply to the COVID-19 pandemic were also presented to participants in parentheses after most items. Participants rated the degree to which their university played a role in the COVID-19 pandemic by: not taking proactive steps to prevent COVID-19 transmission or enact safety protocols (e.g., failing to establish or enforce adequate safety and social distancing protocols); creating an environment in which COVID-19 transmission or safety protocol violations seemed common or normal? (e.g., emphasizing low transmission or fatality rates among college students); creating an environment in which COVID-19 transmission and safety protocol violations seemed more likely to occur (e.g., lack of communication among university officials, lack of clear or consistent safety protocols, lack of proper safety equipment or testing); making it difficult to share your concerns about COVID-19 or report a safety violation (e.g., difficulty contacting university leaders or officials, not being given a chance to ask questions or express

**Table 1. Demographic characteristics for a) Study 1 and b) Study 2.**

| a) Study 1 | | b) Study 2 | |
|---|---|---|---|
| **Gender Identity** | **n (% of 309)** | **Gender Identity** | **n (% of 283)** |
| Woman | 221 (71.5) | Woman | 196 (69.3) |
| Man | 82 (26.5) | Man | 81 (28.6) |
| Non-Binary/Non-Conforming/Not Listed | 6 (1.9) | Non-Binary/Non-Conforming/Not Listed | 6 (2.1) |
| **Race** | | **Race** | |
| American Indian/Native American | 3 (1.0) | American Indian/Native American | 1 (0.4) |
| Asian/Asian American | 23 (7.4) | Asian/Asian American | 18 (6.4) |
| Black/African American | 11 (3.6) | Black/African American | 5 (1.8) |
| Biracial/Multiracial | 34 (11.0) | Biracial/Multiracial | 36 (12.7) |
| Native Hawaiian/Pacific Islander | 1 (0.3) | Native Hawaiian/Pacific Islander | 1 (0.4) |
| White/European American | 233 (75.4) | White/European American | 220 (77.8) |
| Not listed here/Prefer to self-describe/No answer | 1 (0.3) | Not listed here/Prefer to self-describe/No answer | 2 (0.7) |
| **Ethnicity** | | **Ethnicity** | |
| Hispanic/Latino | 50 (16.2) | Hispanic/Latino | 51 (18.0) |
| Non-Hispanic/Latino | 258 (83.5) | Non-Hispanic/Latino | 231 (81.6) |
| No answer | 1 (0.3) | No answer | 1 (0.4) |
| **Student Status** | | **Student Status** | |
| First year | 126 (40.8) | First year | 144 (50.9) |
| Second year | 82 (26.5) | Second year | 79 (27.9) |
| Third year | 69 (22.3) | Third year | 40 (14.1) |
| Fourth year | 21 (6.8) | Fourth year | 14 (4.9) |
| Other/No answer | 11 (3.6) | Other/No answer | 6 (2.1) |
| **Housing** | | **Housing** | |
| Living in on-campus housing | 108 (35.0) | Living in on-campus housing | 114 (40.3) |
| Not living in on-campus housing | 201 (65.0) | Not living in on-campus housing | 169 (59.7) |
| **City of Residence** | | **City of Residence** | |
| Living in university town full time | 187 (60.5) | Living in university town full time | 172 (60.8) |
| Not living in university town full time | 122 (39.5) | Not living in university town full time | 111 (39.2) |
| **In-person classes** | | **In-person classes** | |
| 1+ in-person class | 40 (12.9) | 1+ in-person class | 25 (8.8) |
| No in-person class | 269 (87.1) | No in-person class | 258 (91.2) |
| **COVID-19 exposure** | | **COVID-19 exposure (self)** | |
| Know someone who has contracted C-19 | 224 (72.5) | Tested positive for C-19 | 47 (16.6) |
| Do not know anyone who has contracted C-19 | 84 (27.2) | Have not tested positive for C-19 | 236 (83.4) |
| No answer | 1 (0.3) | **COVID-19 exposure (close other)** | |
| | | Close other tested positive for C-19 | 145 (51.3) |
| | | Close other has not tested positive for C-19 | 138 (48.7) |

concerns); responding inadequately to your concerns about COVID-19 or reports of safety violations (e.g., you were given incorrect or inadequate information or advice that was not feasible for you to follow, your concerns were minimized or invalidated); mishandled a complaint or report related to COVID-19 safety protocols (e.g., failed to adequately investigate or follow procedures); covering up instances of COVID-19 transmission or safety protocol violations (e.g., failure to publicly report accurate COVID-19 transmission rates, failure to inform

students of potential COVID-19 exposure); denying your experience in some way (e.g., your concerns about safety were treated as invalid, your pre-existing condition was dismissed as unimportant); punishing you in some way for expressing concerns about COVID-19 transmission or safety protocol violations (e.g., taking away privileges, being reprimanded); suggesting your experience might affect the reputation of the institution (e.g., suggesting that success of the institution was more important than following COVID-19 guidelines, emphasizing the financial situation of the institution); creating an environment where you no longer felt like a valued member of the institution (e.g., feeling as though the institution does not care about your safety or health); and creating an environment where continued membership was difficult for you (e.g., continued access to your education was financially or personally difficult without support from the institution). "Yes" responses were coded as 1 and were summed to create a total IBQ score ranging from 0 to 12. The distribution was positively skewed (1.39) and kurtotic (1.34), but within the range in which the assumption of normality can be maintained without transformation.

**Current trauma symptoms.** Current trauma-related symptoms were measured using the Trauma Symptoms Checklist (TSC-40) [17], which is a widely used measure of various symptoms related to traumatic experiences. The scale consists of several subscales, including the Dissociation subscale, Sleep Disturbance subscale, Sexual Problems subscale, Anxiety subscale, Depression subscale, and Sexual Abuse Trauma index subscale. For the present study, only the total overall TSC score was used for analysis, and items were summed and averaged to create an average TSC score for each participant. Participants were asked to rate the frequency of experiencing various symptoms (e.g., headaches, depressed mood, anxious thoughts) within the past two months, using anchors ranging from 0 ("Never") to 3 ("Often"). The scale demonstrated satisfactory reliability in this current study ($\alpha$ = .94). The distribution of TSC scores were within a normal range (*skew* = 0.66, *kurtosis* = 0.06).

**COVID-19-specific trauma cognitions.** COVID-19-specific trauma cognitions were measured using an altered version of the Impact of Events Scale (IES) [15] that has been adapted to apply to COVID-19 [18]. This scale measures intrusion and avoidance cognitions related to COVID-19. Intrusion cognitions (e.g., "I had trouble falling asleep because thoughts about COVID-19 came into my mind") and avoidance cognitions (e.g., "I avoided letting myself get upset when I thought about COVID-19 or was reminded of it") often occur following exposure to a specific traumatic event. Participants were asked to rate the frequency of each symptom in the past week, using anchors ranging from 0 ("Never") to 3 ("Often"). Items were summed and averaged to create an average IES score for each participant. The scale demonstrated satisfactory reliability in this current study ($\alpha$ = .90). The distribution of IES scores were within a normal range (*skew* = 0.37, *kurtosis* = -0.49).

**Demographic information.** Participants answered several questions about their demographic information, including age, gender, and race/ethnicity. Students also reported their year/academic status at the university, whether they were living in on-campus university housing, whether they were currently residing in the university town, whether they were currently enrolled in in-person classes, and whether they knew someone personally who had contracted COVID-19.

## Procedure

All study procedures were approved by the University of Oregon Office of Research Compliance (Institutional Review Board). In the present study, participants reviewed an informed consent form before participation. Participants were required to indicate that the read, understood, and agreed to the information presented in the informed consent form by clicking

"Agree" at the bottom of the form. Due to the online nature of the study, a waiver of written consent was obtained. After consenting to participate, participants completed questionnaires through an online survey hosted on Qualtrics survey software from a personal electronic device in a private location of their choosing. Participants had the option to leave items blank and to discontinue participation at any time without penalty. Upon completion of the survey, participants were provided with a debriefing form and received course credit for their participation.

## Data preparation & analysis plan

**Statistical software.** We used *R* (Version 4.0.2) [19] for our analyses. When cleaning data and conducting our analyses, we used the following *R* packages: *dplyr* (Version 1.0.4) [20], *ggplot2* (Version 3.3.2) [21], *psych* (Version 2.0.7) [22], *performance* (Version 0.7.0) [23], and *tidyverse* (Version 1.3.0) [24].

**Missing data.** Out of 23,484 individual data points used in the final analysis (prior to calculation of any average index scores), 62 were missing (0.26%). No individual item had a missing rate higher than 1.3% (4 missing responses out of 309 participants). Due to the low rate of missing data, we opted to not impute missing data. For participants who completed >80% of the items on the TSC and IES, average scores were calculated across completed items (also known as available item analysis) [25]. Although details are not reported in this manuscript, we re-ran analyses using listwise deletion across all variables, and results and statistical inferences did not significantly differ.

**Outlier analysis.** We assessed both TSC and IES scores for outliers (defined as 1.5 x the interquartile range of the respective distribution). Two outliers were identified on the TSC, and zero outliers were identified on the IES. Outliers were capped at values corresponding to lower or upper 5% of the respective distributions. Although two outliers were identified on the IBQ, we opted not to cap these scores because these scores were from a history measure (as opposed to a psychological construct measure) and thus could well be representative of those students' experiences with the university. Although details are not reported in this manuscript, analyses were run both with and without outlier procedures on all variables, and results did not significantly differ on any of our analyses.

**Statistical inference.** Inferential statistics were interpreted using the standard significance threshold ($p < .05$) with two-tailed statistical tests.

## Study 1 results

The majority of students (67.0%; $n = 207$) reported at least one type of COVID-19-related institutional betrayal. The most common types of institutional betrayal reported were "creating an environment in which COVID-19 transmission was more common or seemed normal" and "creating an environment in which COVID-19 transmission seemed more likely to occur" (Fig 1A). There were no significant differences in COVID-19-related institutional betrayal: across genders, $F(2, 302) = 2.18$, $p = .11$; by enrollment in in-person classes, $t(303) = 0.40$, $p = .69$; by residence in on-campus housing, $t(303) = 0.22$, $p = .83$; residence in the university town, $t(303) = 2.50$, $p = .11$; or knowing someone with COVID-19, $t(302) = 3.37$, $p = .07$. There were no significant differences in institutional betrayal by race or gender.

In order to determine the associations between COVID-19-related institutional betrayal and our two outcomes of interest, Pearson's *r* correlation coefficients were calculated. Institutional betrayal ratings were significantly associated with both current trauma symptoms (Fig 2A) and COVID-19 specific intrusion and avoidance cognitions (Fig 2B), $p < .001$ (Table 2).

In order to determine the unique relationship between COVID-19-related institutional betrayal and our two outcomes of interest, we calculated two multiple regression models,

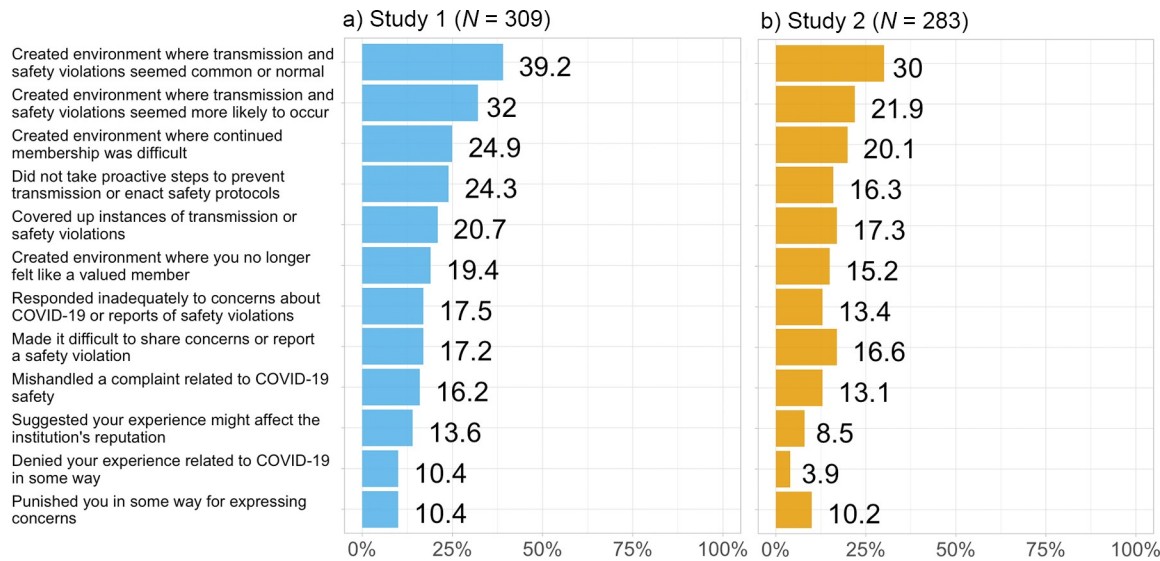

**Fig 1.** Percentage of participants in a) Study 1 and b) Study 2 endorsing each type of COVID-19-related institutional betrayal.

controlling for covariates related to trauma symptoms (gender and familiarity with someone infected with COVID-19). Prior to running each model, we examined model assumptions (e.g., multicollinearity, normality of residuals, homoscedasticity, and homogeneity of observations), and the model appeared to conform to the necessary assumptions of multiple regression. Institutional betrayal was associated with unique variance in current trauma symptoms, even when controlling for gender and knowing someone with COVID-19, $p = .002$ (see Table 3A). Institutional betrayal was also associated with unique variance in intrusion and avoidance cognitions, $p = .004$ (see Table 3B).

## Study 1 discussion

This study is the first to investigate institutional betrayal in undergraduate students' experiences at their university institution during the COVID-19 pandemic. We found that students are experiencing institutional betrayal related to their universities' handling of COVID-19-related safety concerns. The most common instances of institutional betrayal (e.g., "creating an environment where transmission and/or safety violations seemed common or normal" and "creating an environment where transmission and/or safety violations seemed more likely to occur") are also the most common types reported in prior studies on institutional betrayal following sexual assault and harassment [4, 12]. Punishment for reporting and active denial of students' experiences (forms of institutional commission) were the least commonly reported in this study, yet still reported by 10% of students in our sample, which is a concerning statistic.

Contrary to our first hypothesis, there were no significant differences in rates of COVID-19-related institutional betrayal by students' enrollment in in-person classes, students' residence in on-campus housing, or students' residence in the university town. There are many possible reasons for the lack of significant differences. It may be that, in fact, safety protocols are being properly implemented in in-person classes and in dormitories, such that these students do not experience elevated levels of institutional betrayal in these specific domains. Alternatively, students who are not residing close to campus and are operating from an entirely remote position may experience additional forms of institutional betrayal and

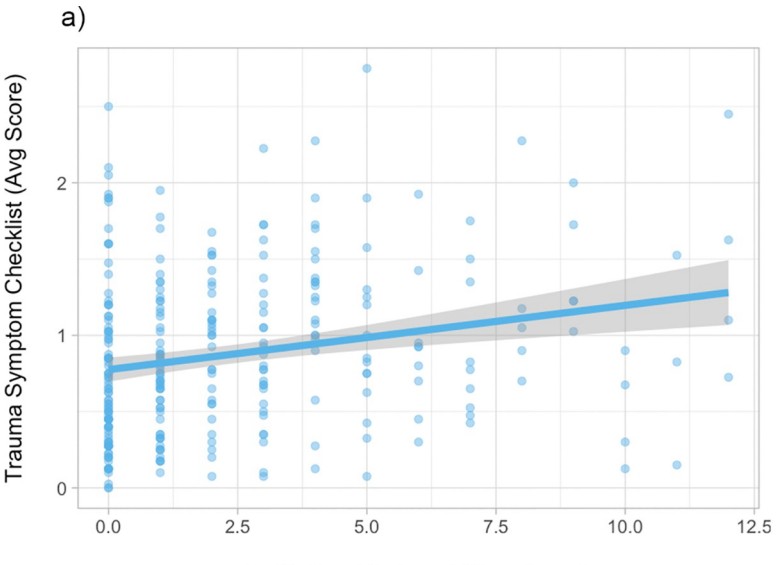

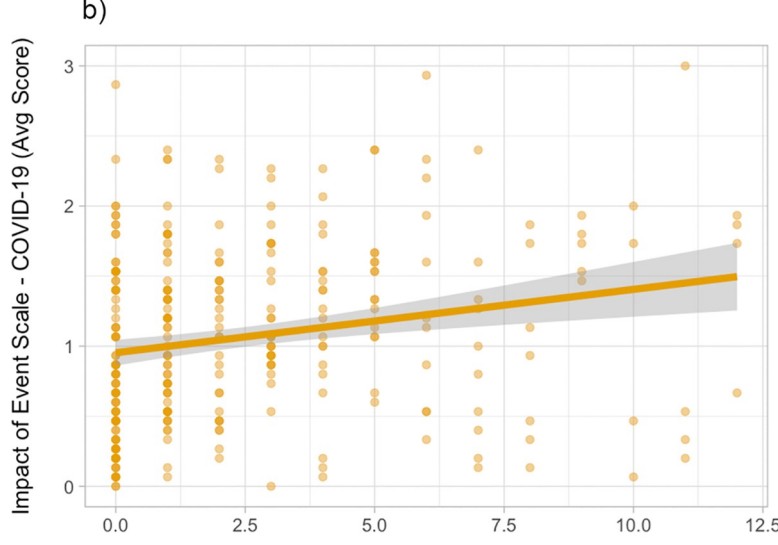

**Fig 2.** The relationship between COVID-19-related institutional betrayal and a) TSC scores and b) IES scores in Study 1 ($N = 309$).

**Table 2. Means, standard deviations, and correlations of variables of interest in Study 1 ($N = 309$).**

| Variable | M | SD | 1 | 2 |
|---|---|---|---|---|
| 1. Institutional betrayal | 2.46 | 2.89 | | |
| 2. Trauma symptoms | 0.88 | 0.52 | .20*** [.09, .31] | |
| 3. COVID-19 cognitions | 1.07 | 0.63 | .20*** [.09, .31] | .48*** [.38, .56] |

***$p < .001$

**Table 3. Current trauma symptoms and COVID-19 cognitions predicted by gender, exposure to someone infected with COVID-19, and COVID-19 institutional betrayal ($N = 309$).**

| Predictor | b | SE | β | t | Fit |
|---|---|---|---|---|---|
| **a) Trauma Symptoms (TSC)** | | | | | |
| Intercept | 0.52 | 0.07 | – | – | |
| Woman | 0.33 | 0.06 | 0.29 | 5.25*** | |
| Non Binary/Non-Conforming | 0.81 | 0.21 | 0.22 | 3.94*** | |
| COVID-19 Exposure–Other | 0.03 | 0.06 | 0.03 | 0.56 | |
| COVID-19 Institutional Betrayal | 0.03 | 0.01 | 0.17 | 3.11** | |
| | | | | | $R^2 = .14$*** |
| **b) COVID-19 Cognitions (IES)** | | | | | |
| Intercept | 0.52 | 0.09 | – | – | |
| Woman | 0.39 | 0.08 | 0.30 | 5.12*** | |
| Non Binary/Non-Conforming | 0.69 | 0.25 | 0.15 | 2.77** | |
| COVID-19 Exposure–Other | 0.23 | 0.08 | 0.16 | 3.05** | |
| COVID-19 Institutional Betrayal | 0.03 | 0.01 | 0.16 | 2.94** | |
| | | | | | $R^2 = .15$*** |

**$p < .01$
***$p < .001$

disconnection from the institution that are less likely to be experienced by students living on or near campus. As such, although the types of institutional betrayal may look different based on these factors, the overall level of institutional betrayal may not be significantly different. Future research should investigate this trend.

Our second hypothesis was supported. Similar to prior research on institutional betrayal in different contexts [4, 7, 11], we found that experiences of institutional betrayal are associated with symptoms of general trauma symptoms (e.g., depression, anxiety, sleep problems). We also found that experiences of institutional betrayal are associated with COVID-19-specific intrusion and avoidance cognitions (e.g., intrusive thoughts about COVID-19, spending a lot of time trying to avoid thinking about COVID-19). The relationships between COVID-19-related institutional betrayal, current trauma symptoms, and COVID-19-specific cognitions persisted, even when controlling for gender and familiarity with someone infected with COVID-19. Similar unique associations between institutional betrayal and symptoms of distress have been replicated in numerous other studies [4, 11, 12].

A limitation of Study 1 was that we did not measure participants' prior trauma history Therefore, we were unable to determine how much variance in trauma symptoms and trauma cognitions might be accounted for by other traumatic experiences and the degree to which institutional betrayal would predict these outcomes above and beyond these past experiences. Furthermore, we designed the initial study and began collecting data during a time in which the COVID-19 pandemic was relatively contained on the university campus, and only a small minority of the campus community had tested positive for COVID-19. Thus, we did not ask participants in the study whether they themselves had tested positive for COVID-19, as it was unlikely that we would obtain sufficient numbers of participants to conduct any meaningful comparisons. Instead, we only asked students if they personally knew another person who had been infected with COVID-19. However, as the academic year progressed and COVID-19 infections rapidly increased among the campus and national communities, personal COVID-19 infection, as well as infection among close family members and friends, become more likely for our participants. Such variables would likely influence experiences of COVID-19-related

institutional betrayal and current distress symptoms. Because of these limitations, we ran an additional study (Study 2) during the winter 2021 academic quarter to replicate Study 1 and extend our research to include the influence of key confounding variables.

## Study 2

Study 2 served to investigate the relationships found in Study 1 during the following academic quarter, during which COVID-19 infections spread more rapidly. Study 2 also aimed to extend the results of Study 1 by controlling for additional covariates (trauma history, personal and familial COVID-19 infection exposure) that may influence the relationships of interest. We had three main research goals:

1. Determine if the rates of COVID-19-related institutional betrayal found in Study 2 would be similar to rates found in Study 1.

2. Test the hypothesis that institutional betrayal would be similarly associated with current trauma symptoms and COVID-19-specific intrusion and avoidance cognitions.

3. Test the hypothesis that experiences of COVID-19-related institutional betrayal would be related to both current trauma-related symptoms and COVID-19-specific intrusion and avoidance cognitions, even when controlling for gender, COVID-19 infection among self, COVID-19 infection among close family or friends, and prior trauma history.

## Study 2 methods

### Participants

Participants were recruited from the same university Human Subjects Pool described in Study 1. A total of 324 undergraduate students signed up and consented to participate in Study 2. Similar to Study 1, participants who failed to correctly answer at least four out of five "attention check" questions randomly located throughout the survey (e.g., "please choose 'strongly agree' if you are paying attention") were removed prior to data analysis ($n$ = 41). The final sample used for analysis consisted of 283 participants (69.3% women, 28.6% men, 2.1% transgender/ non-binary/gender-nonconforming). The majority of participants were White (77.8%) and first-year (50.9%) or second-year (27.9%) students. The average age of participants was 19.41 ($SD$ = 2.21). Demographic characteristics of this sample were similar to Study 1. Full demographic characteristics for Study 2 are listed in Table 1B.

Study 2 data were collected during the winter 2021 quarter of the academic year (February 10 –March 24, 2021). There were no significant changes in COVID-19 policy during the winter 2021 quarter, and the university continued to adopt largely remote instruction model. Like Study 1, this study was part of a larger data collection project, which included other measures of individual differences (e.g., personality characteristics, social attitudes) that were not examined for this study's research questions.

### Measures

**COVID-19-related institutional betrayal.** COVID-19-related institutional betrayal was measured using the adapted version of the Institutional Betrayal Questionnaire-12 (IBQ; see Study 1 description) [4, 12].

**Current trauma symptoms.** Current trauma symptoms were measured using the Trauma Symptoms Checklist-40 (TSC-40; see Study 1 description) [17]. In this study, the scale

demonstrated satisfactory reliability ($\alpha$ = .94), and the distribution of TSC scores was approximately normal (*skew* = 0.41, *kurtosis* = -0.19).

**COVID-19-specific trauma cognitions.** COVID-19-specific intrusion and avoidance cognitions were measured using an adapted version of the Impact of Events Scale (IES) [15], specific to COVID-19 (see Study 1 description) [18]. In this study, the scale demonstrated satisfactory reliability ($\alpha$ = .88), and the distribution of IES score was approximately normal (skew = 0.22, kurtosis = -0.85).

**Trauma history.** Participants' trauma history was measured using the Brief Betrayal Trauma Scale (BBTS) [26]. The BBTS is a widely used measure of trauma that assess participants' exposure to a wide variety of traumatic events that range in the degree of interpersonal betrayal involved. Participants were asked to rate the frequency with which they have experienced 14 traumatic events during both childhood (prior to the age of 18) and adulthood (after the age of 18), with response options including "Not at all" (coded as 0), "1–2 times" (coded as 1), and "More than that" (coded as 2). The 14 traumatic events on the BBTS fall into two general categories–low betrayal or high betrayal. Low betrayal traumatic events involve no interpersonal component (e.g., natural disasters, accidents) or are perpetrated by someone unknown to the victim (e.g., attack by a stranger). High betrayal traumatic events are perpetrated by someone close to and trusted by the victim (e.g., sexual abuse by a caregiver). Participants' coded responses were summed to create separate total scores for both low betrayal trauma and high betrayal trauma history.

**COVID-19 exposure (self and close other).** Participants were asked if they had ever tested positive for COVID-19. Participants who indicated that they tested positive were coded as "1," and participants who denied ever testing positive were coded as "0." Participants were asked if they knew someone personally who had contracted COVID-19, and if applicable, they were asked to think about the person with whom they have the closest relationship who has tested positive. They were then asked describe the relationship, with response options including: "close family member," "close friend," extended family member," "friend," "neighbor," "acquaintance/classmate," or "distant acquaintance/stranger." Participants who indicated that they knew a close family member or close friend were coded as "1," and participants who did not indicate a close family member or friend were coded as "0."

**Demographic information.** Similar to Study 1, demographic information was collected regarding each participant's age, gender, and race/ethnicity. Students also reported their year/ academic status at the university, whether they were living in on-campus university housing, whether they were currently residing in the university town, and whether they were currently enrolled in in-person classes.

## Procedure

The procedure of Study 2 closely mirrors the procedure of Study 1. All study procedures were approved by the University of Oregon Office of Research Compliance (Institutional Review Board). Participants reviewed an informed consent and were required to indicate that the read, understood, and agreed to the information presented in the informed consent form by clicking "Agree" at the bottom of the form. Due to the online nature of the study, a waiver of written consent was obtained. Participants then completed questionnaires through an online Qualtrics survey. Participants received course credit for their participation. All study procedures were approved by the university's Office of Research Compliance.

## Data preparation & analysis plan

**Statistical software.** Similar to Study 1, we used *R* (Version 4.0.2) [19] for our analyses. When cleaning data and conducting our analyses, we used the following *R* packages: *dplyr*

(Version 1.0.4) [20], *ggplot2* (Version 3.3.2) [21], *psych* (Version 2.0.7) [22], *performance* (Version 0.7.0) [23], and *tidyverse* (Version 1.3.0) [24].

**Missing data.**   Out of 30,564 individual data points used in the final analysis (prior to calculation of any average index scores), 363 were missing (1.2%). Due to the low rate of missing data, we opted to not impute missing data. The majority of the missing data points were due to a technical error on the presentation of the BBTS items for the first 20 participants in the survey. A participant notified us of the error, and changes were made to the Qualtrics survey that prevented the error from occurring in additional data collection. These first 20 participants were not included in analyses using the BBTS, and they did not differ significantly from the other participants on IBQ, TSC, or IES scores. Outside of the first 20 participants who viewed an incorrectly formatted version of the BBTS, no individual item had a missing rate higher than 1.1% (3 missing responses out of 283 participants). For participants who completed >80% of the items on the TSC and IES, average scores were calculated across completed items (also known as available item analysis) [25]. Although details are not reported in this manuscript, we re-ran analyses using listwise deletion across all variables, and results and statistical inferences did not significantly differ.

**Outlier analysis.**   As in Study 1, we assessed both TSC and IES scores for outliers (defined as 1.5 x the interquartile range of the respective distribution). Three outliers were identified on the TSC, and zero outliers were identified on the IES. Outliers were capped at values corresponding to lower or upper 5% of the respective distributions. Although two outliers were identified on the IBQ, we opted not to cap these scores because these scores were from a history measure (as opposed to a psychological construct measure) and thus could well be representative of those students' experiences with the university. Although details are not reported in this manuscript, analyses were run both with and without outlier procedures on all variables, and results did not significantly differ on any of our analyses.

**Statistical inference.**   Inferential statistics were interpreted using the standard significance threshold ($p < .05$) with two-tailed statistical tests.

## Study 2 results

The majority of students (54.8%; $n = 155$) reported at least one type of COVID-19-related institutional betrayal. Like Study 1, rates of institutional betrayal did not vary by living on campus, living in the university town, taking in-person classes, gender, or race. The rate COVID-19 institutional betrayal was significantly lower than the rate (67.0%) found in Study 1, $\chi^2(1) = 8.78$, $p = .003$. Total IBQ scores also similarly differed from Study 1 to Study 2, $t(590) = 2.62$, $p = .009$. However, the most common types of institutional betrayal found in Study 1 were replicated in Study 2 (Fig 1B). Pearson's $r$ correlation coefficients were calculated between COVID-19-related institutional betrayal and our outcomes of interest. Institutional betrayal was significantly associated with both current trauma symptoms and COVID-19-specific intrusion and avoidance cognitions, $p < .001$ (Table 4). The correlation between institutional betrayal and trauma symptoms found in Study 2 did not significantly differ from the correlation found in Study 1, Fisher's $z = 0.13$, $p = .90$. The correlation between institutional betrayal and intrusion and avoidance symptoms found in Study 2 also did not significantly differ from the correlation found in Study 1, Fisher's $z = 0.13$, $p = .90$.

In order to determine the unique relationship between COVID-19-related institutional betrayal and our two outcomes of interest, we calculated two multiple regression models, controlling for gender, low betrayal trauma history, high betrayal trauma history, and COVID-19 exposure (self and close other). Prior to running each model, we examined model assumptions (e.g., multicollinearity, normality of residuals, homoscedasticity, and homogeneity of

**Table 4. Means, standard deviations, and correlations of variables of interest in Study 2 (N = 283).**

| Variable | M | SD | 1 | 2 | 3 | 4 |
|---|---|---|---|---|---|---|
| 1. Institutional betrayal | 1.87 | 2.56 | | | | |
| 2. Trauma symptoms | 0.95 | 0.50 | .21*** [.09, .31] | | | |
| 3. COVID-19 cognitions | 0.99 | 0.57 | .19** [.08, .30] | .38*** [.28, .48] | | |
| 4. Low betrayal trauma | 2.48 | 2.95 | .13* [.01, .25] | .33*** [.22, .44] | .16* [.03, .28] | |
| 5. High betrayal trauma | 2.83 | 3.56 | .11^ [-.01, .23] | .38*** [.27, .48] | .06 [-.06, .18] | .57*** [.48, .65] |

^p < .10

*p < .05

** p < .01

***p < .001.

observations), and the models appeared to conform to the necessary assumptions of multiple regression. Institutional betrayal was associated with unique variance in current trauma symptoms, even when controlling for our covariates, $p$ = .03 (see Table 5). However, institutional betrayal was not associated with unique variance in intrusion and avoidance cognitions to a statistically significant degree, $p$ = .08 (see Table 5).

**Table 5. Current trauma symptoms and COVID-19 cognitions predicted by gender, COVID-19 infection (self and close other), trauma history, and COVID-19 institutional betrayal (N = 263).**

| Predictor | b | SE | β | t | Fit |
|---|---|---|---|---|---|
| **a) Trauma Symptoms (TSC)** | | | | | |
| Intercept | 0.47 | 0.07 | – | – | |
| Woman | 0.30 | 0.06 | 0.28 | 4.81*** | |
| Non-Binary/Non-Conforming | 0.29 | 0.21 | 0.08 | 1.36 | |
| COVID-19 Exposure–Self | -0.06 | 0.08 | -0.04 | -0.75 | |
| COVID-19 Exposure–Close Other | 0.01 | 0.06 | 0.01 | 0.17 | |
| Low Betrayal Trauma History | 0.03 | 0.01 | 0.18 | 1.56* | |
| High Betrayal Trauma History | 0.03 | 0.01 | 0.24 | 3.30** | |
| COVID-19 Institutional Betrayal | 0.02 | 0.01 | 0.12 | 2.12* | |
| | | | | | $R^2$ = .26*** |
| **b) COVID-19 Cognitions (IES)** | | | | | |
| Intercept | 0.52 | 0.08 | – | – | |
| Woman | 0.35 | 0.08 | 0.29 | 4.62*** | |
| Non Binary/Non-Conforming | 0.17 | 0.26 | 0.04 | 0.67 | |
| COVID-19 Exposure–Self | 0.09 | 0.10 | 0.06 | 0.93 | |
| COVID-19 Exposure–Close Other | 0.22 | 0.07 | 0.19 | 3.16** | |
| Low Betrayal Trauma History | 0.03 | 0.01 | 0.18 | 2.44* | |
| High Betrayal Trauma History | -0.01 | 0.01 | -0.07 | -0.98 | |
| COVID-19 Institutional Betrayal | 0.02 | 0.01 | 0.11 | 1.72^ | |
| | | | | | $R^2$ = .16*** |

^p < .10

*p < .05

** p < .01

***p < .001. N = 263 (rather than 283) due to error in BBTS presentation (see Methods section for description).

## Study 2 discussion

Study 2 largely replicated the findings from Study 1. In winter quarter 2021, students continued to endorse experiences of COVID-19-related institutional betrayal at high rates, and the most prominent types of institutional betrayal continued to be forms of institutional omission (e.g., "creating an environment where transmission and/or safety violations seemed common or normal" and "creating an environment where transmission and/or safety violations seemed more likely to occur").

Although more than half of students endorsed experiences related to institutional betrayal in both studies, and the distribution of institutional betrayal types was largely consistent across studies, fewer students in Study 2 endorsed experiencing institutional betrayal than in Study 1. This lack of difference is surprising, given that the data from Study 2 was collected during an academic quarter in which there were objectively higher rates of COVID-19 transmission both locally and nationally. Perhaps students may have reported significantly higher rates of COVID-19 institutional betrayal in the fall 2020 quarter (versus winter 2021) because many of these policies and procedures were new and may have been perceived as particularly unsettling or insufficient. By winter 2021, many students may have acclimated to these policies, and both a changing political climate and the initiation of vaccine distribution on the national level may have reduced the perception of institutional betrayal. Alternatively, many of the flawed policies that were initiated in fall 2020 may have been revised, leading to reduced experiences of institutional betrayal. However, the rates of institutional betrayal found in both Study 1 and Study 2 are in line with–if not higher than–rates of institutional betrayal found in prior research, which range from 12% [27] to 66% [5].

Similar to Study 1, COVID-19-related institutional betrayal was similarly correlated with both trauma symptoms and COVID-19-specific intrusion and avoidance cognitions. Unlike Study 1, Study 2 incorporated additional covariates (e.g., trauma history, self and close other COVID-19 infection) when estimating the unique association between institutional betrayal and our two outcomes of interest. Results of these models partially supported our second hypothesis. When accounting for these additional covariates, institutional betrayal explained unique variance in general trauma symptoms, but not intrusion and avoidance.

There may be several explanations for the null finding regarding intrusion and avoidance cognitions. When looking at the standardized $\beta$ value of the regression coefficient, institutional betrayal appears to be positively related to the outcome in a direction consistent with the other results, yet the magnitude of effect did not reach statistical significance, given its small size, the study sample size, our more conservative, two-tailed hypothesis threshold. Despite lack of statistical significance, the direction of the relationship is not incompatible with our other findings. Alternatively, such a relationship between institutional betrayal and intrusion and avoidance symptoms may have existed in the beginning of the pandemic, yet it was diluted as reported rates of institutional betrayal declined from fall to winter.

## General discussion

Results from both Study 1 and Study 2 indicate that students are experiencing institutional betrayal related to their university institution's handling of COVID-19, and this institutional betrayal is related to distress, even when accounting for other covariates. These results have numerous important implications for both students and university administrators alike. First, these results bolster the idea that experiences of trauma-related distress are not solely individual phenomena, but instead are influenced by institutional systems and policies [2, 3]. Although all undergraduate students are currently in the midst of an extraordinarily stressful global

pandemic, institutions, such as universities, have the potential to contribute to and exacerbate students' negative outcomes related to these experiences if they commit institutional betrayal.

The influence of institutional betrayal may have important implications for universities, even after the COVID-19 pandemic is contained. It is inevitable that there will be more pandemics and other major crises in the future. If institutional betrayal becomes the default response throughout this pandemic (and future crises), this may not only influence students' academic performance and identification with the university as a whole. From a pragmatic standpoint, it may also influence students' engagement with university activities, future enrollment, and/or future financial contributions to the university. If universities want to fulfill their promises and continue to provide a community that supports students in the long term, avoiding institutional betrayal in any context is an important first step.

The conclusions garnered from these two studies should be interpreted in light of their limitations. First, both sets of data were collected cross-sectionally, and thus we cannot make causal claims. Given that all measures were collected via self-report, conclusions are also limited by common method variance [28]. Additional, longitudinal data using multiple methodologies will need to be collected in order to establish a direct causal link between students' institutional betrayal experiences and subsequent distress. However, we hope that the establishment of these preliminary relationships will persuade university administrators that students notice institutional betrayal in a variety of contexts and are likely not immune to its noxious effects.

Other notable limitations constrain the generalizability of these results. The data from these studies are situated within the context of a single university in one physical location. As such, it is difficult to conclude how these results generalize to other universities, which may have implemented markedly different COVID-19 policies and procedures. We hypothesize that universities that have implemented stricter policies that prioritize students' health and well-being over financial gain may be less at risk of committing institutional betrayal, and these students may be experiencing less psychological distress. However, such a conclusion is beyond the scope of the current study. Further, the university at the center of this study is located in an area of the Northwest United States with limited racial/ethnic diversity. It is unknown how experiences of racism and/or other types of marginalization may influence perceptions of COVID-19-related institutional betrayal at the university level.

Future research should build upon this study's limitations. In addition to collecting data longitudinally in a more diverse sample at campuses with varying policies, future research would also benefit from collecting qualitative interview data about students' various experiences of institutional betrayal in their own words throughout the COVID-19 pandemic. (For example, at the authors' institution, an anonymous website not affiliated with the university [29] was created for students to share concerns about COVID-19 policy violations on campus and/or experiences with lack of enforcement from the university. This website was linked to an Instagram account that published these statements from students. At the time of manuscript preparation, this account had 3,532 followers and had posted 137 times).

Additional research should also investigate not only the presence and absence of institutional betrayal, but also of positive and effective responses to the COVID-19 pandemic. Such supportive institutional actions–known as *institutional courage* [30]–are conceptualized not only as the absence of harmful responses, but the intentional incorporation of strategies and policies that center the needs of students, despite consequences for the larger institution and its leaders. In the face of COVID-19 pandemic, courageous institutional actions could have manifested as radical transparency of COVID-19 infection rates or incorporation of students' voices and feedback into COVID-19 policy and enforcement procedures.

We hope that this research serves as an initial step to investigate the prevalence of institutional betrayal in a variety of domains, as well as the role that institutional responses play in predicting individuals' mental health and physical health outcomes following chronic stress and trauma. The goal is not only to acknowledge the harm of and eliminate institutional betrayal, but to replace it with actions that center the needs of its institutional members. During COVID-19 and other future crises, universities may not be able to exert control on a national level, but they may be able to create and enact institutional policies imbued with courage.

## Author Contributions

**Conceptualization:** Alexis A. Adams-Clark, Jennifer J. Freyd.

**Data curation:** Alexis A. Adams-Clark.

**Formal analysis:** Alexis A. Adams-Clark.

**Methodology:** Alexis A. Adams-Clark, Jennifer J. Freyd.

**Project administration:** Alexis A. Adams-Clark.

**Supervision:** Jennifer J. Freyd.

**Visualization:** Alexis A. Adams-Clark.

**Writing – original draft:** Alexis A. Adams-Clark.

**Writing – review & editing:** Alexis A. Adams-Clark, Jennifer J. Freyd.

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
