## [Decision Letter · Decision Letter 0]

5 Jul 2021

PONE-D-21-10525

COVID-19-related institutional betrayal associated with trauma symptoms among undergraduate students

PLOS ONE

Dear Dr. Adams-Clark,

Thank you for submitting your manuscript to PLOS ONE. After careful consideration, we feel that it has merit but does not fully meet PLOS ONE’s publication criteria as it currently stands. Therefore, we invite you to submit a revised version of the manuscript that addresses the points raised during the review process.

We look forward to receiving your revised manuscript.

Kind regards,

Vedat Sar, M.D.

Academic Editor

PLOS ONE

Journal Requirements:

**2. **Please include your full ethics statement in the ‘Methods’ section of your manuscript file. In your statement, please include the full name of the IRB or ethics committee who approved or waived your study, as well as whether or not you obtained informed written or verbal consent. If consent was waived for your study, please include this information in your statement as well.

Reviewers' comments:

Reviewer's Responses to Questions

**Comments to the Author**

1. Is the manuscript technically sound, and do the data support the conclusions?

Reviewer #1: Yes

Reviewer #2: Yes

2. Has the statistical analysis been performed appropriately and rigorously? 

Reviewer #1: Yes

Reviewer #2: Yes

3. Have the authors made all data underlying the findings in their manuscript fully available?

Reviewer #1: Yes

Reviewer #2: Yes

4. Is the manuscript presented in an intelligible fashion and written in standard English?

Reviewer #1: Yes

Reviewer #2: Yes

5. Review Comments to the Author

Reviewer #1: This study (actually two studies) represents original research that has not been published elsewhere.

The Introduction adequately covers the available literature and demonstrates a good grasp of such literature.

The Methods sections for both studies, including the description of the study samples, measurements, data management and statistical analyses, are described in sufficient detail and demonstrate a good grasp of quantitative research methodology. Ethical aspects are adequately addressed.

The Results of both studies are presented adequately.

The Discussion of both studies interprets the results adequately in the light of the available literature. The conclusions of both studies are supported by the data.

The presentation of the article is neat, thorough and intelligible. The article adheres to appropriate reporting guidelines and standards for data availability.

The article makes an important original contribution to the existing literature on both institutional betrayal-related research and COVID-19-related research.

Reviewer #2: The manuscript, “COVID-19-related institutional betrayal associated with trauma symptoms among undergraduate students” presents two studies, both employing cross-sectional survey designs, to examine the prevalence of experiences of COVID-19 institutional betrayal among college undergraduates and to test hypotheses about associations between college students’ circumstances, experiences of institutional betrayal, and reported trauma symptoms. Findings suggest that over half of students report at least one type of COVID-19 institutional betrayal by their university and that COVID-19 institutional betrayal is associated with trauma symptoms and COVID-specific trauma cognitions. This paper is well-written and covers an important and timely topic. However, I had some concerns about conceptualization of COVID-19 institutional betrayal, the sample, and analyses. I’ve noted these concerns and others by major section of the paper below in hope that they may be helpful to the authors in refining their paper.

Introduction

• Although the authors provide a definition of institutional betrayal, it might also be helpful to provide a definition of what would count (and not count) as COVID-19 institutional betrayal. Some of the examples provided seemed rather broad. For example, the authors note that “Even those students at universities that have implemented strict, remote- only instruction may experience a sense of institutional betrayal regarding challenges related to remote learning and academic difficulties, which are exacerbated further by existing inequities.” While challenges of remote learning are certainly related to COVID-19, they seem quite different from things the university does that puts students at risk of COVID-19 infection.

• On page 5, I think “incidence rates of 12 types of COVID-19-related institutional betrayal” should be “prevalence rates of 12 types of COVID-19-related institutional betrayal” as the authors do not have data on new cases of institutional betrayal.

• Stronger theoretical justification could be provided for the a priori hypotheses tested in the two studies. For example, why would certain circumstances (e.g., living in a university town, knowing someone infected with COVID-19) be expected to be associated with institutional betrayal ratings? Additionally, although the studies examine intrusion and avoidance cognitions and make specific hypotheses about their associations with institutional betrayal, these cognitions are not defined or discussed in the introduction.

Studies 1 & 2

• I appreciated the detailed demographics provided on the Study 1 and 2 samples. Given that this samples were drawn from a Human Subjects Pool, it would be helpful to also include information about the extent to which the sample reflects the larger student population of students at the institution.

• In initially describing the COVID-19 institutional betrayal measure, it might be helpful to note that the wording of all items appears in Figure 1. When I reviewed the items, some of them seemed specific to COVID-19 (e.g., Created environment where transmission and safety violations seemed common or normal) but others seemed rather broad and could reflect COVID-19 related issues but also more general issues (e.g., Created environment where continued membership was difficult, Created environment where you no longer felt like a valued member). This speaks to my comments about how COVID-19 institutional betrayal is conceptualized in the study. This could be strengthened so that readers can more clearly understand your operationalization.

• Given that results did not differ without outlier procedures in either study, why was the decision made to cap outliers on the TSC?

• In both studies, I was curious why the authors did not control for race/ethnicity in their multiple regression analyses given that systemic racism can lead to disparities in experiences of trauma.

General Discussion

• The authors do a nice job presenting potential limitations of their study. It might be useful to also discuss potential issues with common method variance as both COVID-19 institutional betrayal and trauma symptoms are collected via self-report.

6. PLOS authors have the option to publish the peer review history of their article (what does this mean?). If published, this will include your full peer review and any attached files.

Reviewer #1: No

Reviewer #2: **Yes: **Jennifer Watling Neal

---

## [Author Response · Author response to Decision Letter 0]

13 Jul 2021

Please see attached response to reviewers word document.

---

## [Decision Letter · Decision Letter 1]

16 Sep 2021

PONE-D-21-10525R1

COVID-19-related institutional betrayal associated with trauma symptoms among undergraduate students

PLOS ONE

Dear Dr. Adams-Clark,

Thank you for submitting your manuscript to PLOS ONE. After careful consideration, we feel that it has merit but does not fully meet PLOS ONE’s publication criteria as it currently stands. Therefore, we invite you to submit a revised version of the manuscript that addresses the points raised during the review process.

We look forward to receiving your revised manuscript.

Kind regards,

Vedat Sar, M.D.

Academic Editor

PLOS ONE

Journal Requirements:

Reviewers' comments:

Reviewer's Responses to Questions

**Comments to the Author**

1. If the authors have adequately addressed your comments raised in a previous round of review and you feel that this manuscript is now acceptable for publication, you may indicate that here to bypass the “Comments to the Author” section, enter your conflict of interest statement in the “Confidential to Editor” section, and submit your "Accept" recommendation.

Reviewer #2: (No Response)

2. Is the manuscript technically sound, and do the data support the conclusions?

Reviewer #2: Yes

3. Has the statistical analysis been performed appropriately and rigorously? 

Reviewer #2: Yes

4. Have the authors made all data underlying the findings in their manuscript fully available?

Reviewer #2: Yes

5. Is the manuscript presented in an intelligible fashion and written in standard English?

Reviewer #2: Yes

6. Review Comments to the Author

Reviewer #2: Thanks for the opportunity to re-review the manuscript, “COVID-19-related institutional betrayal associated with trauma symptoms among undergraduate students” which was resubmitted to PLOS-ONE. In this round of revisions, the authors were very responsive to my earlier concerns about conceptualization of COVID-19 institutional betrayal, the sample, and analyses. I think this paper makes an important contribution to the literature on institutional betrayal and have just one remaining minor comment (see below):

• I appreciated the additional context about the extent to which study samples reflect the larger population of students at the institution. Would it be possible to add information about the student population to Table 1 so that readers can directly compare sample to population demographics? This might require some reformatting to get everything to fit but I think this could easily be done if the variable names were only listed once in the table.

7. PLOS authors have the option to publish the peer review history of their article (what does this mean?). If published, this will include your full peer review and any attached files.

Reviewer #2: **Yes: **Jennifer Watling Neal

---

## [Author Response · Author response to Decision Letter 1]

16 Sep 2021

Thank you for taking the time to review our paper entitled “COVID-19-related institutional betrayal associated with trauma symptoms among undergraduate students” for publication in PLOSONE. We appreciate the reviewers’ careful consideration of this article and their receptiveness to our revision. The requested revision by the reviewer was to include demographic statistics for the university community in Table 1. Rather than include it in Table 1, we have decided to instead include additional information in the text of the Methods section regarding specific demographic characteristics of the university student body with citation. This decision was made because publicly available data from the university conflates race and ethnicity, which we separate in our reporting of demographics. They also made several other decisions in the presentation of their data, such as excluding international students, that does not align with our approach. We list these caveats along with the new information we included. In the process of including this information, we also realized that our ethnicity data was missing from Table 1. We included that information in Table 1, so that readers could compare it more readily to the university demographics. We believe these revisions have strengthened the manuscript.

---

## [Editor Report · Decision Letter 2]

24 Sep 2021

COVID-19-related institutional betrayal associated with trauma symptoms among undergraduate students

PONE-D-21-10525R2

Dear Dr. Adams-Clark,

We’re pleased to inform you that your manuscript has been judged scientifically suitable for publication and will be formally accepted for publication once it meets all outstanding technical requirements.

Kind regards,

Vedat Sar, M.D.

Academic Editor

PLOS ONE
---

## [Editor Report · Acceptance letter]

29 Sep 2021

PONE-D-21-10525R2 

COVID-19-related institutional betrayal associated with trauma symptoms among undergraduate students 

Dear Dr. Adams-Clark:

I'm pleased to inform you that your manuscript has been deemed suitable for publication in PLOS ONE. Congratulations! Your manuscript is now with our production department. 

Kind regards, 

on behalf of

Dr. Vedat Sar 

Academic Editor

PLOS ONE